# A Novel Framework of Manifold Learning Cascade-Clustering for the Informative Frame Selection

**DOI:** 10.3390/diagnostics13061151

**Published:** 2023-03-17

**Authors:** Lei Zhang, Linjie Wu, Liangzhuang Wei, Haitao Wu, Yandan Lin

**Affiliations:** 1Academy for Engineering and Technology, Fudan University, Handan 220, Shanghai 200433, China; 2ENT Institute and Otorhinolaryngology Department, Eye & ENT Hospital of Fudan University, Shanghai 200433, China; 3School of Information Science and Technology, Fudan University, Handan 220, Shanghai 200433, China

**Keywords:** unsupervised learning scheme, manifold learning, deep convolutional neural networks, laryngoscopic images, informative frame selection

## Abstract

Narrow band imaging is an established non-invasive tool used for the early detection of laryngeal cancer in surveillance examinations. Most images produced from the examination are useless, such as blurred, specular reflection, and underexposed. Removing the uninformative frames is vital to improve detection accuracy and speed up computer-aided diagnosis. It often takes a lot of time for the physician to manually inspect the informative frames. This issue is commonly addressed by a classifier with task-specific categories of the uninformative frames. However, the definition of the uninformative categories is ambiguous, and tedious labeling still cannot be avoided. Here, we show that a novel unsupervised scheme is comparable to the current benchmarks on the dataset of NBI-InfFrames. We extract feature embedding using a vanilla neural network (VGG16) and introduce a new dimensionality reduction method called UMAP that distinguishes the feature embedding in the lower-dimensional space. Along with the proposed automatic cluster labeling algorithm and cost function in Bayesian optimization, the proposed method coupled with UMAP achieves state-of-the-art performance. It outperforms the baseline by 12% absolute. The overall median recall of the proposed method is currently the highest, 96%. Our results demonstrate the effectiveness of the proposed scheme and the robustness of detecting the informative frames. It also suggests the patterns embedded in the data help develop flexible algorithms that do not require manual labeling.

## 1. Introduction

Laryngeal cancer (LC), grouped with head and neck squamous cell cancer (HNSCC), is the 7th most common cancer (men 5th and women 13th) [1]. In Europe, it is the second most common malignancy of the head and neck region [2]. A landmark report confirmed the evidence of the association between tobacco smoke and cancer in the 1950s, explicitly in the head and neck tumors, including the larynx [3,4]. Nearly 87% of LC patients are tobacco users (central Europe), while 60–89% of LC is attributed to a combination of tobacco smoking and alcohol drinking (South America). Fortunately, quitting cigarettes lowers the probability of developing laryngeal cancer, especially for those who had smoked for more than ten years [5]. The Surveillance, Epidemiology, and End Results (SEER) also reported new laryngeal cancer cases fell by some 50% from 1975 to 2016, credited to lower smoking rates in younger populations and the evolving tobacco-related marketing [1].

The 5-year survival rate is one of the most concerning indicators in the larynx community for computer-aided diagnosis (CADx) [6,7]. It notes the percentage of people still alive more than five years after confirmed laryngeal cancer [8]. In recent cancer statistics, laryngeal cancer is not one of the leading cancers in the United States [9]. Unfortunately, the 5-year survival rate has dropped from 66 to 63% in the past 40 years [10], while approximately 60% of patients present with advanced stage disease (stage III or IV) at diagnosis [11].

### 1.1. Early Diagnosis and Narrow-Band Imaging

Nowadays, surgical techniques performed for laryngeal function preservation are feasible. However, the early diagnosis of laryngeal cancer is still the primary means of clinical intervention. Ref. [12] showed that patients with different levels of laryngeal carcinomas (Tis, T1, and T2) have an 80–90% probability of healing in the early stage, during an approximately 60% cure rate for more advanced tumors. Ref. [4] calls on governments and social institutions to get involved in cancer prevention education, early diagnosis, surveillance, and monitoring of public health interventions.

The current diagnosis of LC at the early stage is not out of the scope of an endoscopy. The literature has well summarized the pros and cons of the two popular non-invasive endoscopy techniques, narrow-band imaging (NBI) and white light endoscopy (WLE). NBI is more expensive than WLE from a clinical perspective. However, it is beneficial for adding more definition to the tumor margins and highlighting the features of submucosal vascularization [7,13]. Additionally, ref. [14] reported that NBI is 21% more diagnostically sensitive than conventional WLE.

### 1.2. Informative Frame Selection for CADx

Manually selecting the informative frames from the vast number of candidates available in endoscopy is tedious and time-consuming. It also requires experienced endoscopists. More than 10,000 endoscopy frames are produced per patient during each examination. It may cost the physician a lot of time (over 60 min) to read the images [15,16]. Not all frames are helpful; most are redundant (blurred, bubbles, etc.), and only a tiny proportion are related to lesions or abnormalities. Therefore, selecting the most informative instances (images) is essential for the subsequent automatic diagnosis, such as classification and segmentation of the lesions [17,18]. If the uninformative frames are removed, the accuracy of the automatic detection of the abnormalities will be increased [19].

This issue seemed to be trivial with the emergence of deep learning techniques. However, we examined several recent studies that employed artificial neural networks for the computer-aided diagnosis of laryngeal cancer [7,20,21,22,23] and found that although the datasets were labelled and well-structured, the number of images still ranged from 3000 to 25,000. Such samplings applied to clinical applications are still a considerable workload for science, technology, engineering, and math (STEM) researchers willing to cooperate with physicians [24].

On the other hand, collecting high-quality data is a critical research component, ultimately improving laryngeal cancer patient management [1]. Interestingly, the public datasets related to informative frame selection are rare, only 3% (3/97) compared to vibration analysis, lesion recognition, etc. [24]. Our fundamental concern is that existing algorithms are insufficient for extracting specific patterns of high-quality data in enormous data with noise. The solution for informative frame selection should be close to a natural and clinical setting, with unstructured and unlabeled data. Ref. [18] reported a similar scenario in the field of biomedicine and biology.

### 1.3. Organization

The remaining paper is organized as follows. Section 2 illustrates the related works and the missing part of the puzzle. Section 3 describes the proposed approach to select the informative frames in an unsupervised fashion, a scheme combined with feature embedding, dimensionality reduction methods, and the primary clustering methods. Section 4 presents the experimental details, including the dataset and metrics. It includes the automatic labeling algorithm and a cost function for hyperparameter tuning. Results are presented in Section 5 and discussed in Section 6. Finally, the paper concludes with suggestions in Section 7.

**Figure 1 diagnostics-13-01151-f001:**
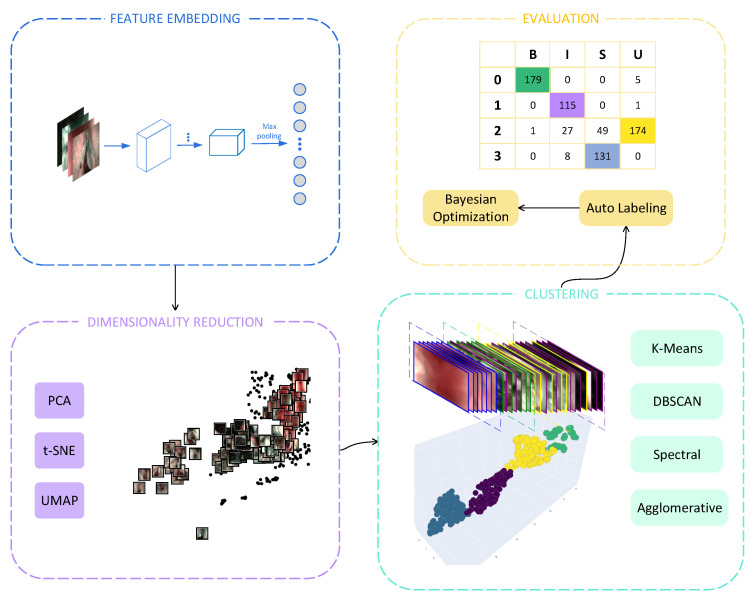
The flowchart scheme.

## 2. Related Work

Many medical screenings include informative frame selection, such as image screening for laryngeal disease detection and redundant image elimination. These are not imited to laryngoscopy [17,25], e.g., gastrointestinal endoscopy [26], wireless capsule endoscopy (WCE) [27,28], optical endomicroscopy [29], and colonoscopy [30]. We have grouped these methods into two categories.

### 2.1. Criterion-Based Feature Extraction

This method’s main characteristic is based on observing basic information about the image, such as color, texture, and geometric features. The statement is that the sharp images respond to high-frequency content [31].

Ref. [26] found that the informative and uninformative frames can be distinguished in the frequency domain according to an energy histogram. Later, the k-means algorithm was employed in the endoscopic video manifolds to cluster the informative frames. However, the frequency spectrum contains superfluous information. Ref. [17] selected a set of descriptors followed by a support vector machine to classify the NBI endoscopic frames. Their method overcame the specific threshold. Similarly, ref. [17] observed that informative frames have higher spatial frequencies than blurred ones. Refs. [27,32] extracted the features with a local color histogram in the HSV space and isolated non-informative frames with a support vector machine classifier. Refs. [25,33] employed the Shannon entropy to eliminate the uninformative images in their specific endoscopic applications. Ref. [29] grouped pure noise and motion artifacts into uninformative frames of optical endomicroscopy. They used grey level cooccurrence matrix (GLCM) texture metrics, such as contrast, energy, entropy, etc., to detect the uninformative frame in the video sequences. Ref. [34] proposed four criteria for screening sharp laryngoscopic images, including sum-modulus-differences and the energy Laplacian of the image, gradient magnitude maximization, and variance. In their latest study, ref. [35] analyzed the hue and geometric features of the laryngoscopic images and introduced the peak signal-to-noise ratio (PSNR) to calculate the nearest frames in the video.

### 2.2. Learning-Based Feature Extraction

The criterion-based methods focus on distinguishing informative frames with criterion functions and are subjective and time-consuming. However, learning-based approaches extract and represent features without any explicit method [16]. Such methods find representations directly from the input frames without a specific feature set or descriptors. Ref. [36] constructed two convolutional neural networks to detect the blurry and water frames from colonoscopy videos.

Recently, several studies have adapted the idea of transfer learning, one of the emerging techniques in deep learning. Ref. [37] employed the pre-trained Inception-v3 network to end-to-end classify the informative and non-informative frames from a colonoscopy. Ref. [38] used the fine-tuned VGG16 networks to classify the different categories of the NBI frames from laryngoscopic videos. Ref. [39] evaluated several state-of-the-art convolutional neural networks in the NBI-InfFrames dataset and achieved state-of-the-art performance. As of this writing, a study proposed a dataset to evaluate the pre-trained ResNet-18 model. The dataset contains 22,000 laryngoscopic frames. It is the second dataset announced for the problem of informative frame selection and the largest one [40].

Therefore, no matter what feature extraction methods are employed, considerable research has been dedicated to employing supervised learning methods as the classifier due to the types of uninformative frames defined. However, we cannot easily employ such methods to detect uninformative frames for unknown types in clinical setting data. A revolutionary study categorized the frames of WCE videos into four representative groups without defining any frame types [28]. They employed non-negative matrix factorization (NMF) and fuzzy C-means (FCM) to eliminate 85% of uninformative frames. However, they did not take the cluster number determination into account.

### 2.3. Contributions

We try to answer whether informative frame selection can be conducted without supervision or guidance from the labels. We hypothesised that informative frame selection does not rely on data labels. Our proposed unsupervised framework (Figure 1) remains competitive compared to the state-of-the-art methods in the NBI-InfFrames dataset. The main contributions of this paper are summarized as follows:A new scheme couples the state-of-the-art dimensionality reduction techniques and clustering methods for solving the issue. The proposed scheme extracts feature faster than the baseline method [17] and require no effort on the labeling data. This ensures the reliability and effectiveness of the proposed scheme in a clinical setting datasets.For the frame reduction or video summarization, the future direction for unsupervised learning methods should involve cluster determination [28]. In this work, we introduce a metric under our scheme, the Calinski-Harabasz Index, to automatically determine cluster numbers.In this work, we propose an automatic cluster labeling algorithm using bijections mapping for evaluating the classification performance of unsupervised methods. We further propose a Bayesian optimization cost function algorithm to boost classification performance.To the best of the authors’ knowledge, none of the existing works in the literature on computer-aided diagnosis of laryngoscopy attempt to solve the problem using an unsupervised scheme based on the feature learning method. In addition, our methods achieved comparable performance to state-of-the-art supervised learning methods [17,38,39].

## 3. Methods

The unsupervised learning algorithm is a typical process to find patterns in data without labels [41]. It is usually used for clustering, feature extraction, or dimensionality reduction [42].

### 3.1. Feature Embedding

Two categories of feature extraction can be summarized: the methods that rely on conventional machine learning and those that rely on neural networks. Ref. [16] illustrated more specific divisions as spatial domain, frequency domain, and feature learning methods. Considering this work, the dataset we use in Section 4.1 only contains 720 images. Thus, we use a small architecture neural network. Another fact is that ref. [38] employed the fine-tuned VGG16 succeeded on NBI-InfFrames, motivated by transfer learning.

The vanilla VGG16 network without any pre-trained weights is used in this work for feature extraction. Moreover, the last two layers were dropped (Figure 2).

### 3.2. Dimensionality Reduction

The dimensionality reduction technique is not rare in the area of medical imaging. It is widely used for feature selection and visualization. In the area related to endoscopic images, ref. [26] proposed a manifold learning method named EVM for projecting the endoscopic video into the local structure of the manifold, and ref. [28] introduced an unsupervised data reduction algorithm for the capsule endoscopy video segments, the non-negative matrix factorization method. This work introduces the current state-of-the-art techniques to compare with a recently proposed algorithm.

#### 3.2.1. PCA

Principle components analysis (PCA) is a popular dimensionality reduction technique that maps the data points from high- to low-dimensional space with the linear transformation. In mathematical terms, the transformation can be denoted as [43]
(1)Y=XM,
where X represents the original data or features, Y is the matrix of the transformed data points $y_{i}$, and the mapping relationship is represented by M. PCA aims to find the *M* that maximizes the cost function trace $tr(M^{T}cov\left(X\right)M)$, where cov(X) is the sample covariance matrix. Consequently, it is transferred to solve the eigenproblem of the *d* principle eigenvectors (principal components),
(2)cov(X)M=λM.

In the experiment, *d* is decided by analysis of the variance. Nearly 80% of the variance can be explained by the 50 principal components. Thus, we choose d=50 in Section 5.1.

Minimizing the cost function forms of the Euclidean distance to find the *M* is commonly used in multidimensional scaling [43].
(3)$$\phi \left(Y\right)=\sum_{ij}\left(d_{ij}^{2}-{∥y_{i}-y_{j}∥}^{2}\right),$$
where $d_{ij}$ represents the Euclidean distance between the $x_{i}$ and $x_{j}$ in high-dimensional space, and the $∥y_{i}-y_{j}{∥}^{2}$ is the square Euclidean distance between the low-dimensional space.

#### 3.2.2. t-SNE

A stochastic neighbour embedding technique, t-SNE, was presented by Matten and Hinton [44]. It is widely recommended due to its good visualizations. SNE aims to find the optimal mapping relationship between high and low-dimensional space by minimizing the mismatch between $p_{j|i}$ and $q_{j|i}$. As the derivative version of SNE, t-SNE overcomes the drawbacks of using the non-symmetric Kullback–Leibler divergence to measure the faithfulness of the pairwise distance. The Student-t distribution is used to compute the similarity between two data points in the low-dimensional space instead of a Gaussian, thus named t-SNE.

To minimize the cost function of the conditional probabilities $p_{j|i}$ and $q_{j|i}$, t-SNE turns to minimize a single Kullback–Leibler divergence between the joint probability, *P*, in high-dimensional space and a joint point probability, *Q*, in low-dimensional space. Thus, the cost function is denoted as
(4)$$C_{t-SNE}=KL\left(P∥Q\right)=\sum_{i\neq j}\sum_{j}p_{ij}\log\frac{p_{ij}}{q_{ij}},$$
where the low-dimensional pairwise similarity map by $q_{ij}$ and the high-dimensional map $p_{ij}$ are given by
$$p_{ij}=\frac{\exp\left(-{∥x_{i}-x_{j}∥}^{2}/2\sigma^{2}\right)}{\sum_{k\neq l}\exp\left(-{∥x_{k}-x_{l}∥}^{2}/2\sigma^{2}\right)},q_{ij}=\frac{{\left(1+{∥y_{i}-y_{j}∥}^{2}\right)}^{-1}}{\sum_{k\neq l}{\left(1+{∥y_{k}-y_{l}∥}^{2}\right)}^{-1}}.$$

The $p_{ij}$ is a Gaussian distribution to approximate the high-dimensional space, while the $q_{ij}$ is a Student-t distribution. The detailed optimization of the cost function using the stochastic gradient descent is available in Section 3.3.4 of this article [44].

#### 3.2.3. UMAP

Although the t-SNE is good at revealing important global structures, it cannot go beyond locality-preserving limits. Uniform manifold approximation and projection (UMAP) [45] takes a big step in preserving the global structure of the large dataset, as well as the small one. It implies that the distance of the inter-class data points is more distinguishable under the dimensionality reduction techniques with the globality-preserving properties. This is the key to understanding that UMAP outperforms other dimensionality reduction techniques. We will explain this finding further in Section 6.

Our interest in the UMAP came from visualising artworks of the Metropolitan Museum of Art collection  [46]. We observed that the artworks are widely distributed according to the brightness and darkness of the average intensity after projection by UMAP. Also, the distinguishable characteristic is embedded in the NBI-InfFrames dataset. The image thumbnails of the informative video frames are well separated from the non-informative frames (Figure 1 DIMENSIONALITY REDUCTION).

Since the mathematical language to understand the UMAP is similar to the t-SNE, we present this part in Appendix B. It is also reported that the UMAP is nine times faster than the t-SNE as evaluated on the MNIST dataset with the scikit-learn toolkit [47].

### 3.3. Clustering

We divide the clustering methods into four main categories in this work. They are hierarchical methods, partitioning methods, graph-based methods, and density-based methods [18]. Existing clustering methods, such as fuzzy and soft clustering, are not included in the investigation.

#### 3.3.1. Agglomerative

Hierarchical clustering groups the data point into a binary tree called a dendrogram [48]. Agglomerative clustering is one of the two hierarchical clustering methods widely used for its simplicity. Its history dates back at least to the 1950s. Agglomerative clustering builds clusters in a bottom-up fashion, starting with each data point from its cluster. In the subsequent step, the two closest clusters will be merged until all data points are grouped into one cluster. Different from the iterative clustering algorithms, the data points cannot be further merged adjustments or split once the building progress of the agglomerative clustering is made. The property can be helpful for applications with an unknown number of clusters, such as bioinformatics applications [49].

Factors such as similarity measures, criterion functions, and initial conditions, decide the effectiveness of the clustering methods. For agglomerative clustering, there are three definitions of the similarity between two clusters: single-link, complete-link, and average-link [48]. Such linked strategies appear in the form of the hyperparameters (Section 4.4) in our experiments.

#### 3.3.2. K-Means

A partitional clustering method manipulates a single partition on the data points, while the hierarchical clustering method holds a whole structure of the clustering, described as the dendrogram [50]. Therefore, the computation time of the hierarchical clustering performed on the large-size dataset is unbearable. K-means is one of the partitional clustering methods with linear computational time employed in a wide range of applications.

A pattern x is defined as a singleton (observation, feature vector, or datum) on the clustering algorithm, consisting of *d* dimensional measurements $x=\left(x_{1},\ldots x_{d}\right)$. A pattern set is denoted as $X=\{x_{1},\ldots ,x_{n}\}$, which can be viewed as a matrix of samples and features (n×d). The clustering L of the k-means can be established from the pattern set X by employing a squared error criterion as follows
(5)$$e^{2}\left(X,L\right)=\sum_{j=1}^{K}\sum_{i=1}^{n_{j}}{∥x_{i}^{\left(j\right)}-c_{j}∥}^{2},$$
where the $x_{i}^{\left(j\right)}$ is the $i^{th}$ pattern of the $j^{th}$ cluster, and $c_{j}$ is the centroid calculated by the mean of the member points in the $j^{th}$ cluster.

In summary, the algorithm of k-means starts with the initial position and relocates patterns to the clusters according to the similarity measurement until the convergence criterion is satisfied. This study was motivated by the classification performance of the k-means outperforming the baseline work. In the implementation, the MinibatchKMeans was employed due to the faster convergence and low difference in accuracy compared to k-means.

#### 3.3.3. Spectral Clustering

Until now, we can conclude that the goal of the clustering methods is to minimize the differences in the same cluster and maximize the difference between the clusters. Spectral clustering aims to achieve this by partitioning the similarity graph, consisting of the eigenvalues of the similarity matrix of the data.

Spectral clustering is an algorithm that the k-means performed on the eigenvectors of the graph Laplacian [26]. In addition, it can be viewed as a variant kernel k-means overcoming the drawback of k-means in the non-linear space.

Given the undirected graph G=(V,E), it consists of two elements: the vertex $v_{i}\in V$ and the edge $e_{ij}\in E$. Each vertex represents a data point $x_{i}$. If the edge $e_{ij}$ between two points $x_{i}$ and $x_{j}$ is larger than a certain threshold, we denote they are connected and weighted by the similarity $s_{ij}$, thus $s_{ij}\geq 0$.

Assuming the graph *G* is weighted, an adjacency matrix $W={\left(w_{ij}\right)}_{i,j=1,\ldots ,n.}$ is then generated, which describes the similarity between the vertices $v_{i}$ and $v_{j}$. In addition, $w_{ij}=0$ in the adjacency matrix *W* means there is no connection between the two points.

The degree $d_{i}$ of the vertex $v_{i}$ is naturally introduced, as $d_{i}=\sum_{j=1}^{n}w_{ij}$. It counts all the weights starting from the vertex $v_{i}$, referred to as $v_{ij}$, but not including the vertices ending at $v_{i}$, referred to as $v_{ji}$. The degree matrix of *D* is referred to as the diagonal of the matrix with the degree of vectors $d_{1},\ldots ,d_{n}$.

An *A* is denoted as the subset of the vertices *V*, A⊂V; thus, we have the indicator vector $1_{A}={\left(f_{1},\ldots ,f_{n}\right)}^{\prime }\in R^{n}$ [51],
(6)$$f_{i}=\left\{\begin{array}{cc} 1, & \mathrm{if}\;v_{i}\in A\\ 0, & \mathrm{otherwise}\; \end{array}\right..$$

The unnormalized graph Laplacian is generated from the components of the matrix of *D* and the matrix of *W*. The defined matrix can be given as
(7)L=D−W.

Once the graph is constructed, the spectral clustering algorithm can be interpreted to the k-means. Computing the first *k* eigenvectors of the *L* from the equation (Equation (Equation 7)), denoted as the $u_{1},\ldots ,u_{k}$. Using the terms of the k-means, a pattern $U\in R^{n\times k}$, *k* is the column number of the *U*. Later, the clustering $Y=\{y_{i}|i=1,\ldots ,n\}$, the rows *i* represent the samples. Different criterion functions can be employed on the U and Y, such as the Euclidean, Manhattan, cosine, etc. Finally, the algorithm outputs the cluster indices, $A_{i}=\left\{j|y_{j}\in C_{i}\right\}$.

#### 3.3.4. DBSCAN

DBSCAN does not hold the common assumption that clusters can be decided by minimizing and maximizing the difference between intra-clusters and inter-clusters. An essential insight is the density of the points; the density of the points in each cluster is higher than the density of the points outside the cluster, and vice versa. For the outliers, the noise (points) density is lower than regular clusters.

Unlike the traditional clustering methods, which hardly assign each point to a cluster, DBSCAN assigns the probability to each point, one of the density-based clustering methods. It only requires a few input parameters (one input parameter used in [52]) and computational efficiency compared to the other three algorithms (Appendix C). The shape of the clusters made by the partitioning methods is convex, while the hierarchical algorithm is prohibitive for the computational time, especially for the large data size. However, it also brings the inconvenience of finding the intended number of clusters for aligning to the number of the ground-truth classes of the dataset. Therefore, we propose Algorithm 1, which helps to find the optimal result for the classification performance.

The algorithm starts from a radius of neighbours; the shape of the radius is determined by the distance measurements. For example, two points, *p* and *q*, are neighbours in the Manhattan space. It can be denoted as dist(p,q), and the shape of these two points is rectangular. The definition of the neighborhood of a point *p* can be denoted as $N_{Eps}\left(p\right)$,
(8)$$N_{Eps}\left(p\right)=\left\{q\in D|dist\left(p,q\right)\leq Eps\right\}.$$

**Algorithm 1** Cost function for the Bayesian optimization searching**Require:** $S_{rec}$, recall of the method; $S_{pre}$, precision of the method; $S_{params}$, parameter space
**Ensure:** number of clusters K = 4 α←100▹ penalty the label counts deviation β←0.01▹ weight for the impact of the variance **repeat**    **if** method is MinibatchKMeans or Agglomerative or Spectral **then**    $p\leftarrow 0.1\cdot |S_{rec}-S_{pre}|$    $C\leftarrow (1-S_{rec})+p$, s.t. $S_{params}$    **else if** menthod is DBSCAN **then**    count the outlier labels, −1 in the cluster, $N_{out}$    count the number of kinds of labels except for outliers in the cluster, $N_{in}$    calculate the variance of the counts of each group deviate from 180, σ    $C\leftarrow N_{out}+\left(1-\beta \right)\cdot (\alpha \cdot |N_{in}-K\left|\right)+\beta \cdot \sigma$, s.t. $S_{params}$    **end if**   $\min_{S_{params}}$ C. **until** stop-iteration criteria satisfied **return** best parameters in $S_{params}$

Other rules for DBSCAN are maintaining its arbitrary shape to discover the non-convex clusters, such as the density-reachable, density-connected, cluster, and noise  [52]. In our experiment, we only specify the minimum number of points (MinPts) and the radius of the shape Eps.

## 4. Evaluation

The experiments are performed on a Linux platform with a 2.00 Hz CPU, 16 GB Tesla P100-PCIE GPU, and 16 GB RAM. The source code for this work is publicly available at https://github.com/portgasray/UL-IFS-LC (accessed on 1 January 2023). In Section 4.1, we describe the dataset and the definition of the categories. Section 4.2 illustrates the evaluation metrics for analysis results. Section 4.3 presents an algorithm for automatically delivering the intent label to the clusters. Finally, the cost function used for finding the optimal result is described in Section 4.4.

### 4.1. Dataset

NBI-InfFrames is a dataset acquired with the NBI endoscopic system (Olympus Visera Elite S190 video processor and an ENF-VH rhino-laryngo videoscope) with a frame rate of 25 fps and image size of 1920 × 1072 pixels [17]. It is the current known labelled dataset that is access available in the laryngoscopy area.

The dataset contains 720 frames in 4 categories collected from 18 patients affected by laryngeal squamous cell carcinoma (SCC). Each category consists of 180 frames, informative (I), blurred (B), specular reflection (S), and underexposed (U), Table A1. The dataset was manually labelled by three human evaluators and split into three folders according to the following criteria [17]:B, frames should show a homogeneous and widespread blur.I, frames should have adequate exposure and visible blood vessels; they may also present micro-blur and small portions of specular reflections (up to 10 per cent of the image area).S, frames should present bright white/light-green bubbles or blobs, overlapping with at least half of the image area.U, frames should present a high percentage of dark pixels, even though small image portions (up to 10 per cent of the image area) with over or regular exposure are allowed.

Sample images for the four classes are shown in Figure 3. The color intensity bar range from 0 to 255 of the whole dataset is shown at the bottom.

### 4.2. Evaluation Metrics

The following metrics will be used to evaluate the classification performance of the state-of-the-art models and our proposed methods.
(9)$$\mathrm{Precision}\left({\mathrm{Prec}}_{class}\right)=\frac{TP}{TP+FP}$$
(10)$$\mathrm{Sensitivity}\left({\mathrm{Rec}}_{class}\right)=\frac{TP}{TP+FN}$$
(11)$$F1-\mathrm{score}\left(F1_{class}\right)=\frac{2}{\frac{1}{{\mathrm{Rec}}_{class}}+\frac{1}{{\mathrm{Prec}}_{class}}}=\frac{2TP}{2TP+FP+FN}$$

ROC/AUC, $\mathrm{FPR}=\frac{FP}{FP+TN}$ is the x-axis for the false positive rate. $\mathrm{TPR}=\frac{TP}{TP+FN}$ is the y-axis for the true positive rate.

The Calinski–Harabasz Index [53] (Variance Ratio Criterion) is a ratio of the sum of the inter-clusters (between-group) dispersion and the intra-cluster dispersion (within-group) for all clusters.
(12)$$BGSS=\sum_{k=1}^{K}n_{k}\times {∥C_{k}-C∥}^{2}$$

The between-group sum of squares (BGSS) is a weighted sum of squared distances between each cluster centroid and the centroid of the whole dataset, where $C_{k}$ is the centroid of the cluster *k*, *C* is the centroid of the whole dataset, and $n_{k}$ is the number of the data points in the cluster *k*.
(13)$$WGSS_{k}=\sum_{i=1}^{n_{k}}{∥X_{ik}-C_{k}∥}^{2}$$

The within-group sum of squares (WGSS) calculates the distance between the data points and the centroid of the same cluster. $X_{ik}$ is the data points in the cluster *k*, and $C_{k}$ is the centroid of the cluster *k*.
(14)$$\mathrm{Calinski-Harabasz}\;\mathrm{Index}\left(CH\right)=\frac{\frac{BGSS}{K-1}}{\frac{WGSS}{N-K}}=\frac{BGSS}{WGSS}\times \frac{N-K}{K-1}$$

Finally, the Calinski–Harabasz Index is calculated from a ratio sum of BGSS and WGSS, where N is the number of all data points and K is the number of clusters divided by the algorithm, a big score (CH) indicates a well-separated performance.

The silhouette score [54] analyzes the separation distance between clusters, and the number ranges from [−1, 1]. A value close to one means the clusters are far away from each, and vice versa. A negative score [−1, 0] indicates that the data points are assigned to the wrong clusters.
(15)$$\mathrm{Silhouette}\;\mathrm{Score}\left(\mathrm{SC}\right)=\frac{\left(b-a\right)}{max(a,b)},$$
where *a* represents the mean distance between a sample point and other points in the same cluster, while *b* represents the mean distance between a sample point and other points in the nearest cluster.

### 4.3. Automatic Cluster Labeling

Evaluating the performance of a clustering algorithm is not as trivial as counting the precision and recall of a supervised classification algorithm [55]. In this section, we propose an automatic cluster labeling algorithm to close the gap that compares the two different kinds of algorithms.

Since the cluster labels have no intention of understanding the ground-truth classes, we manually assign the meaningful intent label to the clustering result by viewing the images in the cluster, which is time-consuming and tedious. Before diving deep into the algorithm, we need to recap some sets and map theories.

Given a set of *N* unlabeled image instances, which can be denoted as $X=\{x_{1},x_{2},...,x_{N}\}$, the *N* is 720 for the NBI-InfFrames in our experiments. Therefore, the intent classes $X_{G}$, are referred to as $X_{G}=\left\{X_{I},X_{B},X_{S},X_{U}\right\}$. Each class in the NBI-InfFrames can be represented by $X_{G}=\left\{X_{j}=\left\{x_{1},x_{2},...,x_{180}\right\}|j\in \left\{I,B,S,U\right\}\right\}$, where I,B,S,U are the categories of the dataset. There are 180 image instances for each category.

A collection of labelled clusters $X_{C}$ is generated from the clustering methods in the scheme, $X_{C}=\left\{X_{C_{i}}|i\in \left\{0,1,2,3\right\}\right\}$, where *i* refers to the four clusters with the inattentive labels. We can conclude the solution for this problem into a bijections map problem. The key is to find the mapping relationship from the inattentive labels generated from clusters $X_{C}$ to the intent classes $X_{G}$. The mapping function $f:X_{C}\to X_{G}$ can be denoted by
(16)$$\forall X_{j}\in X_{G},\exists !X_{C_{i}}\in X_{C}\mathrm{such}\;\mathrm{that}X_{j}=f\left(X_{C_{i}}\right),$$
where $\exists !X_{C_{i}}$ represents exactly one $X_{C_{i}}$ exists.

The implementation of finding the mapping relationship between cluster pseudo labels and the intent labels is proposed (Algorithm 2). In addition, we visualize one of the possible results of the algorithm (Figure 1 EVALUATION).
**Algorithm 2** Automatic cluster labeling**Require:** $X_{C_{i}}$, images grouped by cluster labels; $X_{G}$, images with meaningful intent labels
**Ensure:** number of clusters is 4
 i←{0,1,2,3}
 j←{I,B,S,U}
 **for** each cluster $X_{C_{i}}$ in $X_{C}$ **do**    **for** each class $X_{j}$ in $X_{G}$ **do**     calculate intersection number of the $X_{C_{i}}$ and $X_{j}$, $n(X_{C_{i}}\cap X_{j})$    **end for**    find the max intersection number of $X_{C_{i}}$ in $X_{G}$, max $\left\{n\left(X_{C_{i}}\cap X_{G}\right)\right\}$    update the mapping relationship, f:i→j **end for** **return**
*f*


### 4.4. Cost Function in Hyperparameter Tuning

Hyperparameter tuning contributes to the performance of the proposed methods, which exceed the state-of-the-art supervised learning algorithms. As we introduced 4 clustering methods and 3 kinds of dimensionality reduction methods, the Cartesian product of the two kinds of methods is 12 possible hyperparameter spaces. The mission of the cost function is to find the optimal combination among these spaces (Algorithm 1).

Bayesian optimization (BO) and random research (RS) are designed for this purpose. Evidence shows BO 100× sample efficiency gains more than RS [56]. Our trial on the task finding optimal average sensitivity demonstrated that the BO is faster than RS with 262 s in the 200 steps evaluation and 840 s in 500 steps evaluation. Meanwhile, the difference in average sensitivity between BO and RS is less than 0.3%.

Several strategies are employed to optimize finding the results of the DBSCAN. For the methods of MinibatchKMeans, agglomerative, and spectral clustering, the proposed cost function aims to find the best sensitivity. Meanwhile, it penalizes a deviation. For the DBSCAN, the cost function is used to find the number of clusters, which must be identical to the ground-truth numbers of the classes.

## 5. Experiments and Results

### 5.1. Comparison of Dimensionality Reduction Methods

With the support of the ground-truth label from the NBI-InfFrames dataset, we visualize the feature embeddings using three different dimensionality reduction methods. The data points of different classes are distinguishable in (b) and (c). In contrast, the points of class *S* and class *U* are overlapped in (a) (Figure 4). We cannot further infer from the visual inspection that the UMAP is the most favourable. However, the t-SNE and UMAP are better at visualization than PCA.

To further evaluate the clustering preference of features projected from PCA, t-SNE, and the UMAP, we introduced the silhouette score (SC) (Equation (Equation 15)) for the quantitative analysis of the methods above. We observed that the original feature embeddings of the data points in class *S* and class *U* are negative (Figure 5a). The situation changed while using the PCA projection. However, classes *B*, *I*, and *U* are partly negative. Meanwhile, three classes surpass the average SC, and one is left behind (Figure 5b). Unlike the first two analyses of the silhouette score, the silhouette analysis of the projected features with t-SNE or UMAP is beyond the average silhouette score (Figure 5c,d). Moreover, the negative part of the classes is much less than the first two (Figure 5). Finally, based on the average silhouette score, the clustering performance of the UMAP projected features is a little better than the feature embeddings projected by the t-SNE (0.5 versus 0.48, average silhouette score).

We can deduce that the cluster performance of the introduced UMAP projected features is most favourable via the visual inspection and silhouette analysis of the introduced dimensionality reduction methods. In the subsequent experiments, we combine the cluster methods and several introduced dimensionality reduction methods and further evaluate their classification performance with precision, recall, and F1-score. Eventually, we want to know whether the UMAP is most promising.

Take a typical clustering method as an illustration. The classification performance of the K-means combined with UMAP is best compared to combine with PCA or t-SNE or without projection (the median precision ${\mathrm{Prec}}_{class}=92\%$, recall ${\mathrm{Rec}}_{class}=94\%$, F1-score ${F1}_{class}=93\%$ with respective smallest IQR=7%,7%,5% are reported in Table 1). Meanwhile, the detection of class *I* is most robust (${\mathrm{Rec}}_{class}=95\%$), too, from the perspective of the task purpose. Such observation also can be found in other experiments (Table 2 and Table 3).

Until now, we can conclude that UMAP is better than PCA or t-SNE or without projection for clustering performance both from qualitative and quantitative analysis perspectives. The classification performance of the clustering methods coupled with UMAP is also better than the alternative dimensionality reduction methods, such as PCA and t-SNE, in terms of precision, recall, and F1-score. Finally, we further compare the different clustering methods combined with UMAP in the next section.

### 5.2. Classification Performance Comparison

Based on the prior observations, the clustering methods coupled with the UMAP projected feature achieved the best performance in clustering and classification. We compared the different clustering methods (Section 3.3) coupled with UMAP under the ROC/AUC curve. The comparison results suggest that the agglomerative clustering method obtained the highest mean AUC score, followed by K-means, spectral clustering, and DBSCAN (AUC = 96%, 95%, 95%, 94%, respectively, reported in Figure 6a). In addition, the agglomerative clustering obtained the best median recall with relatively the smallest IQR among all clustering methods (${\mathrm{Rec}}_{class}=95\%$, IQR=6% in Table 3). Meanwhile, the proposed clustering methods coupled with UMAP are beyond the baseline (Figure 6b).

It is worth noting the spectral clustering failed on the PCA and t-SNE projected features while succeeding in the UMAP case (Table 2). We detailed the classification performance of precision, recall, and F1-score of the proposed clustering methods coupled with different dimensionality clustering methods in Table 1 and Table 2. An exception is that DBSCAN differs from the other methods since one cannot specify the number of clusters in advance. Fortunately, we proposed a cost function in the Bayesian optimization searching algorithm to approach the exact number of clusters.

In this section, we found that agglomerative clustering coupled with the UMAP achieved the best classification performance in all clustering methods, and the proposed methods achieved comparable performance compared to the baseline from the perspective of statistical significance (*p* > 0.1, Wilcoxon signed-rank test).

### 5.3. Comparison with Benchmarks

The agglomerative clustering exceeded the baseline by 12% (Table 4). We were inspired to compare the best method in our scheme with several benchmarks on the NBI-InfFrame [17,38,39].

We can infer from the statistical significance that our method (UMAP + Agglo) is comparable to the performance benchmarks (Figure 7). Our method (UMAP + Agglo) achieves identical performance to the current best benchmark in terms of the complete statistics of recall (*p* = 1, Wilcoxon signed-rank test).

So far, we can conclude the best method in our scheme, the agglomerative clustering coupled with UMAP (UMAP + Agglo), obtained a mean AUC = 96% with a standard deviation (±0.01). For detecting the informative frame (class *I*), the method achieved a 97% mean AUC (Figure 8a). In addition, we illustrated the relative confusion matrix of the method for visualization of the classification results (Figure 8).

### 5.4. Cluster Number Determination

We analyzed the classification performance of the proposed clustering methods, as we knew the types of informative and uninformative frames. The types of frames may not be predictable in actual clinical data. Thus, the definition of the informative frame does not exist. We can hypothesise that the exact number of classes of the NBI-InfFrmae dataset is unknown. In other words, the dataset is not well-classed. We introduced the Calinski-Harabasz Index (Equation (Equation 14)) to determine the optimal number of clusters.

We observed existing results matched the exact number of classes of the NBI-InfFrmae dataset. The Calinski–Harabasz Index score in the column of UMAP indicated the optimal number of clusters is four (Table 5). The result suggested that the proposed scheme can still achieve comparable performance in the situation of unknown labels of the NBI-InfFrame.

## 6. Discussion

The purpose of informative frame selection is to detect the informative frames among all kinds of frames. As the class of the NBI-InfFrame is well-defined in advance, we intentionally hide the labels of the dataset and proposed an unsupervised learning scheme. Finally, we evaluated the classification performance of the proposed methods by unfold the labels.

There are several reasons the agglomerative clustering achieved the best classification performance in our scheme:We extracted the features using a suitable scale convolutional neural network.We employed a persevering global-structure features method, which keeps a minimal number of features for the subsequent clustering.The agglomerative clustering maintained a bottom-up fashion dendrogram, which is efficient for the small dataset.

Since the difference in the classification performance between the clustering methods is tiny, we finally attributed the success of the proposed scheme to the introduced dimensionality reduction method, UMAP. Meanwhile, the proposed cost function for the Bayesian optimization searching algorithm is vital in finding optimal parameters in the vast space.

Still, we cannot infer from the classification results that the learning-based feature exaction is superior to the criterion-based one. Most importantly, we revealed an unsupervised scheme that achieved a comparable classification performance to the supervised learning methods without defining types of frames. However, there are several drawbacks to this work:The introduced metric for cluster number determination needs to be further demonstrated in the dataset close to the clinical setting.The proposed automatic cluster labeling algorithm is conditioned on the number of clusters, which should be identical to the defined number of the class.The cost function in Bayesian optimization aims to find the best average recall of all classes; thus, the time consumption of the searching algorithm is enormous (Appendix C).

## 7. Conclusions

In this work, we developed a novel unsupervised framework, integrated with the neural network and dimensionality reduction methods coupled with clustering methods, that can distinguish the informative and uninformative frames from laryngoscopic videos. An automatic cluster labeling algorithm and a cost function for the Bayesian optimization are proposed to manifest the classification performance of the framework.

Several experiments were conducted. The comparison result of the different dimensionality reduction methods showed that the t-SNE and UMAP are more suitable than PCA in our scheme. Furthermore, the UMAP best fits the 4 clusters with an average silhouette score of 0.5, while the t-SNE is 0.48. The four clustering methods coupled with UMAP all obtained comparable performance to the baseline. The comparison among the four clustering methods coupled with UMAP indicated that agglomerative clustering achieved the best classification performance. An overall median classification recall of 96% among four frame classes was achieved with 12% over the baseline. Informative video frames were classified with a recall of 94%. Such performance is comparable to the current optimal benchmark from the statistical significance perspective. Moreover, the Calinski-Harabasz Index in the t-SNE and UMAP separately indicates the optimal number of the cluster is the same as the class number. This evidence motivates the application of the proposed scheme to vanilla clinical data.

This work stands on the experiments of the NBI-InfFrame; further evaluation experiments on the different datasets can be conducted for the community. In other words, the effort of the methods to improve the clinical data quality is worthwhile but not only devoted to machine learning algorithms with manual labeling.

## Figures and Tables

**Figure 2 diagnostics-13-01151-f002:**
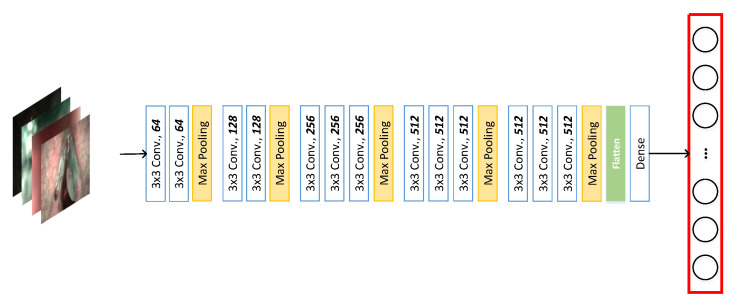
The four images represent the four types of categories in the NBI-InfFrames. The neural network consists of the vanilla VGG16 without the last two layers, the fully connected layer, and the prediction layer. The generated feature embedding dimensionality is 4096.

**Figure 3 diagnostics-13-01151-f003:**
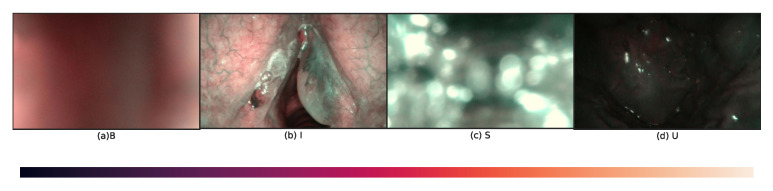
Visualization of sampled examples in the NBI-InfFrames: (**a**) B: blurred frame; (**b**) I: informative frame; (**c**) S: frame with saliva and specular reflections; (**d**) U: underexposed frame. The intensity bar of the dataset is at the bottom.

**Figure 4 diagnostics-13-01151-f004:**
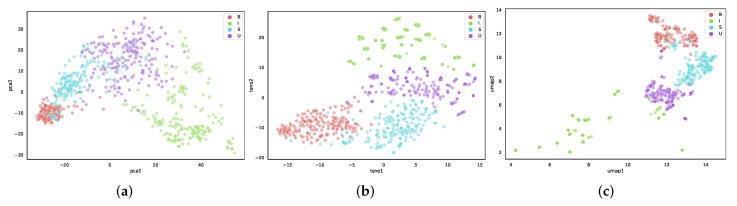
The projections of the feature embedding using different dimensionality reduction methods: (**a**) the original feature embeddings projected by PCA; (**b**) the original feature embeddings projected by t-SNE; (**c**) the original feature embeddings projected by UMAP. The four different frame classes classified by the ground-truth labels are reported (B: blurred frames, I: informative frames, S: frames with saliva or specular reflections, U: underexposed frames).

**Figure 5 diagnostics-13-01151-f005:**
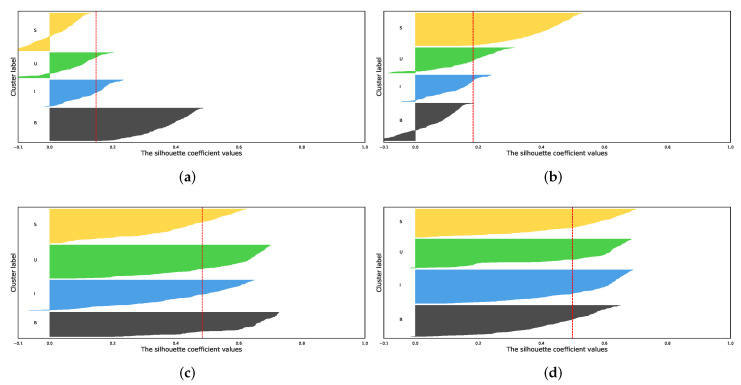
Silhouette analysis for K-means clustering on proposed dimensionality reduction methods. Different frame classes (B: blurred frames, I: informative frames, S: frames with saliva or specular reflections, U: underexposed frames) are in different colors. The red dotted line represents the average silhouette score (avg_sc), and the negative part of the cluster indicates the incorrect clustering. (**a**) silhouette analysis for K-means clustering on vanilla feature embedding (avg_sc = 0.15); (**b**) silhouette analysis for K-means clustering on PCA projected feature embeddings (avg_sc = 0.18); (**c**) silhouette analysis for K-means clustering on t-SNE projected feature embeddings (avg_sc = 0.48); (**d**) Silhouette analysis for K-means clustering on UMAP projected feature embeddings (avg_sc = 0.50).

**Figure 6 diagnostics-13-01151-f006:**
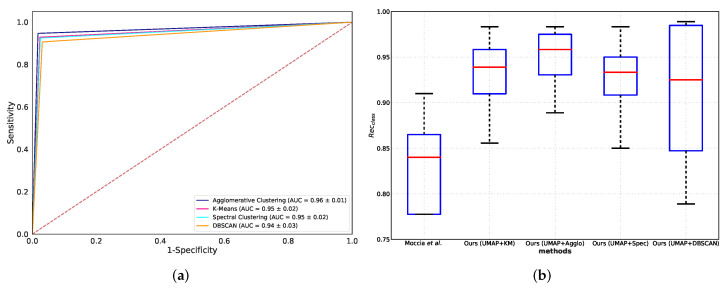
Classification performance comparison of the proposed methods: (**a**) receiver operating characteristic (ROC) curves and area under ROC curve (AUC). The mean area under the ROC curve (±standard deviation) of each method is reported in the legend; (**b**) The boxplot of recall (${\mathrm{Rec}}_{class}$) for comparison of the proposed clustering methods. The comparison in terms of ${\mathrm{Rec}}_{class}$ for the proposed methods and method proposed by [17].

**Figure 7 diagnostics-13-01151-f007:**
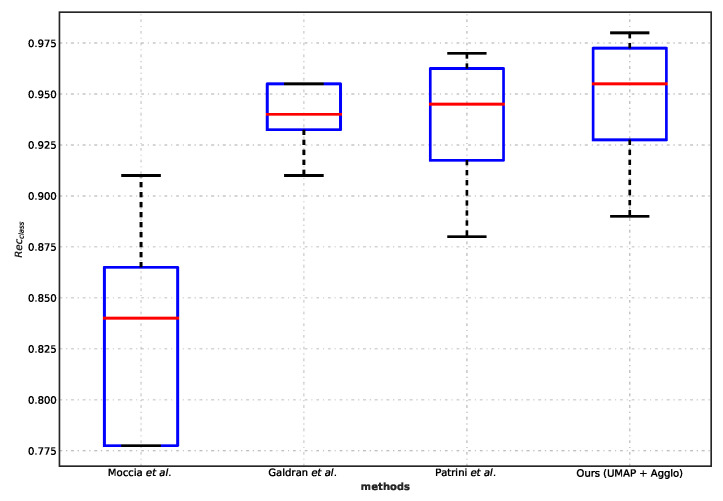
Boxplot of recall (${\mathrm{Rec}}_{class}$) for comparing with benchmark studies. We compared our method (UMAP + Agglo) with [17,38,39] quantitatively using the NBI-InfFrame dataset for evaluation. The difference between the class-specific recall from ours and the other three methods is not statistically significant (relative *p*-value is 0.125, 1.000, 0.625, Wilcoxon signed-rank test). The overall median recall of the proposed method (UMAP + Agglo) outperformed Moccia et al. by 12% absolute.

**Figure 8 diagnostics-13-01151-f008:**
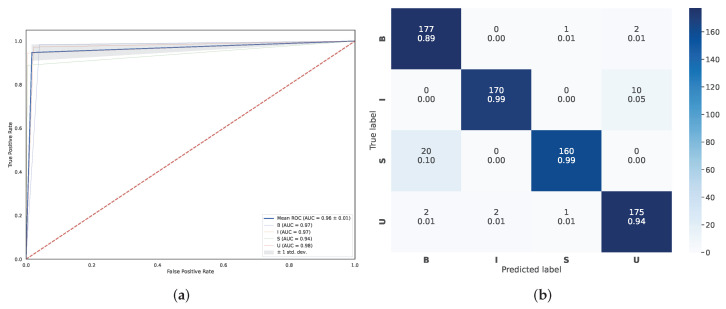
Classification performance of the proposed method (UMAP + Agglo): (**a**) for quantitative analysis, the receiver operating characteristic (ROC) curves and the area under the ROC curve (AUC); the mean (±standard deviation) area under the ROC curve is reported by the solid blue lines (a grey area) in the legend. The area under the ROC (AUC) for each class is reported, too; (**b**) confusion matrix for the proposed method (UMAP + Agglo); the color bar on the right represents the number of frames in each class (B: blurred frames, I: informative frames, S: frames with saliva or specular reflections, U: underexposed frames).

**Table 1 diagnostics-13-01151-t001:** Classification performance of the feature embeddings using the state-of-the-art dimensionality reduction approaches (PCA, t-SNE, UMAP). Results are evaluated under the K-means clustering (KM). Precision (${\mathrm{Prec}}_{class}$), class-specific recall (${\mathrm{Rec}}_{class}$), and F1-score (${F1}_{class}$) are reported for the different frame classes (**B**: blurred frames, **I**: informative frames, **S**: frames with saliva or specular reflections, **U**: underexposed frames) of the NBI-InfFrames dataset. The three metrics’ median and interquartile range (IQR) are also reported.

	Vanilla K-Means			PCA + K-Means			t-SNE + K-Means			UMAP + K-Means		
	${\mathrm{Prec}}_{class}$	${\mathrm{Rec}}_{class}$	${F1}_{class}$	${\mathrm{Prec}}_{class}$	${\mathrm{Rec}}_{class}$	${F1}_{class}$	${\mathrm{Prec}}_{class}$	${\mathrm{Rec}}_{class}$	${F1}_{class}$	${\mathrm{Prec}}_{class}$	${\mathrm{Rec}}_{class}$	${F1}_{class}$
**B**	**0.90**	0.97	0.93	**0.90**	0.97	0.93	0.89	0.97	0.93	0.89	**0.98**	**0.94**
**I**	**1.00**	0.87	0.93	**1.00**	0.87	0.93	**1.00**	0.81	0.89	**1.00**	**0.95**	**0.97**
**S**	**0.94**	0.78	0.85	**0.94**	0.78	0.85	0.88	**0.87**	0.87	0.93	0.86	**0.89**
**U**	0.78	**0.97**	0.87	0.79	**0.97**	0.87	0.80	0.89	0.85	**0.90**	0.93	**0.92**
Median	0.92	0.92	0.90	0.92	0.92	0.90	0.89	0.88	0.88	**0.92**	**0.94**	**0.93**
IQR	0.13	0.15	0.07	0.13	0.15	0.07	0.11	0.09	**0.05**	**0.07**	**0.07**	**0.05**

**Table 2 diagnostics-13-01151-t002:** Classification performance of the feature embeddings using the state-of-the-art dimensionality reduction approaches. Results are evaluated by Spectral clustering (Spec). Precision (${\mathrm{Prec}}_{class}$), class-specific recall (${\mathrm{Rec}}_{class}$), and F1-score (${F1}_{class}$) are reported for the different frame classes (**B**: blurred frames, **I**: informative frames, **S**: frames with saliva or specular reflections, **U**: underexposed frames) of the NBI-InfFrames dataset. The three metrics’ median and interquartile range (IQR) are also reported. The dash in the cells indicates the failure on the PCA projected features and the t-SNE projected features.

	Spectral Clustering			PCA + Spectral Clustering			t-SNE + Spectral Clustering			UMAP + Spectral Clustering		
	${\mathrm{Prec}}_{class}$	${\mathrm{Rec}}_{class}$	${F1}_{class}$	${\mathrm{Prec}}_{class}$	${\mathrm{Rec}}_{class}$	${F1}_{class}$	${\mathrm{Prec}}_{class}$	${\mathrm{Rec}}_{class}$	${F1}_{class}$	${\mathrm{Prec}}_{class}$	${\mathrm{Rec}}_{class}$	${F1}_{class}$
**B**	0.25	1.00	0.40	0.25	1.00	0.40	0.31	1.00	0.47	0.89	0.98	0.93
**I**	0.00	0.00	0.00	0.00	0.00	0.00	1.00	0.74	0.85	0.99	0.94	0.97
**S**	0.00	0.00	0.00	0.00	0.00	0.00	0.00	0.00	0.00	0.92	0.85	0.88
**U**	0.00	0.00	0.00	0.00	0.00	0.00	0.00	0.00	0.00	0.91	0.93	0.92
Median	-	-	-	-	-	-	-	-	-	**0.92**	**0.94**	**0.93**
IQR	-	-	-	-	-	-	-	-	-	**0.06**	**0.07**	**0.05**

**Table 3 diagnostics-13-01151-t003:** Classification performance of the feature embeddings using the state-of-the-art dimensionality reduction approaches. Results are evaluated under agglomerative clustering (Agglo). Precision (${\mathrm{Prec}}_{class}$), class-specific recall (${\mathrm{Rec}}_{class}$), and F1-score (${F1}_{class}$) are reported for the different frame classes (**B**: blurred frames, **I**: informative frames, **S**: frames with saliva or specular reflections, **U**: underexposed frames) of the NBI-InfFrames dataset. The three metrics’ median and interquartile range (IQR) are also reported.

	Agglomerative Clustering			PCA + Agglomerative Clustering			t-SNE + Agglomerative Clustering			UMAP + Agglomerative Clustering		
	${\mathrm{Prec}}_{class}$	${\mathrm{Rec}}_{class}$	${F1}_{class}$	${\mathrm{Prec}}_{class}$	${\mathrm{Rec}}_{class}$	${F1}_{class}$	${\mathrm{Prec}}_{class}$	${\mathrm{Rec}}_{class}$	${F1}_{class}$	${\mathrm{Prec}}_{class}$	${\mathrm{Rec}}_{class}$	${F1}_{class}$
**B**	0.89	0.98	**0.93**	**0.91**	0.94	**0.93**	0.89	**0.99**	**0.93**	0.89	0.98	**0.93**
**I**	**1.00**	0.83	0.91	**1.00**	0.83	0.91	0.95	0.92	0.93	0.99	**0.94**	**0.97**
**S**	**0.99**	0.88	0.93	0.90	**0.91**	0.91	**0.99**	0.60	0.75	**0.99**	0.89	**0.94**
**U**	0.84	**0.99**	0.91	0.84	0.95	0.89	0.71	0.93	0.81	**0.94**	0.97	**0.95**
Median	0.94	0.93	0.92	0.91	0.93	0.91	0.92	0.93	0.87	**0.97**	**0.96**	**0.95**
IQR	0.13	0.13	**0.02**	0.09	0.08	**0.02**	0.17	0.20	0.15	**0.08**	**0.06**	0.03

**Table 4 diagnostics-13-01151-t004:** Classification performance of the state-of-the-art methods and our proposed method. Precision (${\mathrm{Prec}}_{class}$), class-specific recall (${\mathrm{Rec}}_{class}$), and F1-score (${F1}_{class}$) are reported for the four frame classes (**B**, **I**, **S**, **U**). Results from Moccia et al., (2018) [17] proposed SVM with the manually selected feature set, Patrini et al., (2020) [38] proposed VGG16 fine-tuned method, and Galdran et al., (2019) [39] proposed SqueezeNet-based method. The three metrics’ median and interquartile range (IQR) are also reported.

	SVM [17]			Fine-tuned SqueezeNet [39]			Fine-tuned VGG16 [38]			Ours (UMAP + Agglo)		
	${\mathrm{Prec}}_{class}$	${\mathrm{Rec}}_{class}$	${F1}_{class}$	${\mathrm{Prec}}_{class}$	${\mathrm{Rec}}_{class}$	${F1}_{class}$	${\mathrm{Prec}}_{class}$	${\mathrm{Rec}}_{class}$	${F1}_{class}$	${\mathrm{Prec}}_{class}$	${\mathrm{Rec}}_{class}$	${F1}_{class}$
**B**	0.76	0.83	0.79	**0.94**	0.94	0.94	0.92	0.96	0.94	0.89	0.98	0.93
**I**	0.91	0.91	0.91	0.97	**1.00**	0.98	0.97	0.97	0.97	0.99	0.94	0.97
**S**	0.78	0.62	0.69	0.93	**0.91**	0.91	0.93	0.88	0.91	**0.99**	0.89	0.94
**U**	0.76	0.85	0.80	**0.97**	0.94	0.95	0.92	0.93	0.93	0.94	0.97	0.95
Median	0.77	0.84	0.80	0.96	0.94	**0.95**	0.93	0.95	0.94	**0.97**	**0.96**	**0.95**
IQR	0.09	0.16	0.12	0.04	**0.05**	0.04	**0.03**	0.06	0.04	0.08	0.06	**0.03**

**Table 5 diagnostics-13-01151-t005:** Calinski–Harabasz Index (CH) analysis on the cluster number of the K-means coupled with three state-of-the-art dimensionality reduction approaches. Evaluated methods reported without dimensionality reduction (vanilla), PCA, t-SNE, and UMAP. The tested cluster number ranges from 2 to 6.

n_cluster	Vanilla↑	PCA ↑	t-SNE ↑	UMAP ↑
2	**185.55**	**226.95**	1445.76	1097.70
3	148.47	186.72	1470.55	1285.14
4	122.82	157.26	**1696.34**	**1529.70**
5	104.90	136.15	1564.44	1384.12
6	93.23	122.66	1500.42	1357.81

## Data Availability

Data sharing is applicable to this article.

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
