# Peer review of "A Novel Framework of Manifold Learning Cascade-Clustering for the Informative Frame Selection"

_diagnostics, 2023, doi:10.3390/diagnostics13061151_

Round 1

Reviewer 1 Report

In this paper, the authors proposed a computational pipeline for identifying the image of laryngeal cancer. A vanilla neural network combined with UMAP was shown a great performance on the NBI-InfFrames dataset. Although a great deal of work were performed in this study, there are some minor concerns about this research.

Minor:

1.       The detailed description of each method is good, but I didn’t see any detail description of how the datasets were used for model training and testing, especially for the performance comparison. A table with number of data items used for model training and testing (how the items were selected for training and testing) must be added in the manuscript. Is there any cross-validation for the training and testing?

2.       How the ROC curves generated? How many figures were used to generate these ROC curves?

Author Response

A1 for Q1

Thank you for the excellent question. We only employed the vanilla neural network as a feature extractor, as the transfer learning does. In practice, we remove the last two layers. Typically, researchers use the classifier in the neural networks to end2end training a task.

For such a setting, we hypothesise that the informative frame selection can be solved without any labels. As the dataset employed in this work is the NBI-InfFrame, which is labelled by the author. However, we assume there are no labels for this work. Until the proposed method finished the clustering, we employed several techniques to align the cluster numbers with the ground-truth labels, and then the comparison with the supervised-learning methods can be performed. The alignment works can be checked from the algorithms proposed in the article.

The extraction detail can be viewed from here1, the GitHub code by the JupyterNote Book, in the section Feature Extraction by CNN.

A2 for Q2

This question is related to the last one. As your concern, the ROC curves may be a little weird. The technique employed from sk-learn, is the Multiclass Receiver Operating Characteristic.

We generated two curves by the predicted and actual labels on the dataset. The GitHub code can be found here2, in the section of Draw Confusion mx.

As there is no training or test progress for DCNN (vanilla neural network) used in this, we only employed it to extract features from the images. Then we got a 4096-dimensional vector for the following input, the UMAP.

There are two ROC curves. The first one, Fig. 6. It's about the classification performance of the clustering methods, exactly four methods. The second one is the detail of the clustering method, Agglomerative clustering coupled with UMAP (Fig. 8).

Reference

[1,2] https://github.com/portgasray/NCU-AutoHyp/blob/main/inf-sele-1-eval-clustering-on-nbi-inf.ipynb

Thanks for your review and good questions. 

Reviewer 2 Report

1.      Novelty is not clear. Do you present an approach to use detection algorithm in new way or a model of algorithm?

2.      There are no comparisons to other ideas thus we don’t know efficiency of your model.

3.      We don’t know construction of presented model. It is not possible to repeat your calculations for this proposal. Show details of your applied model.

4.      How to set optimal coefficients for this model? Did you test other configurations?

Author Response

A1 to Q1

Yes, it is not very clear for current writing. We failed to write down this hypothesis, leading to the manuscript being misunderstood a lot each time.

We try to answer the question of the informative frame selection that can be solved without any supervision or guidance by the labels. We hypothesised that the informative frame selection does not rely on the data labels. 

Back to the novelty, the aforementioned works, such as the original by Moccia et al. (2018), later Patrini et al. (2019) and so on. They assumed the label was needed, no matter the methods employed, the SVM, and the fine-tuned VGG16. They did not find out that the informative frame selection succeeded without any supervision at that moment.

This work assumes that If we do not know the labels of the dataset, how are things going?

We first extracted the features by the vanilla neural networks and sent the extracted features into the manifold space to reduce the dimensionality. The figure tells us the structure of the data point can be clearly viewed from 2D visualization. It demonstrated that our hypothesis is correct. At last, we only employed unsupervised learning methods, such as clustering, rather than classifiers. The quantitative results showed that the performance of unsupervised learning is comparable to the state-of-the-art methods, even though we compared it with those supervised-learning methods.

In general, the novelty is not about the proposed specific model. We found a scheme/framework step by step following the experiments. The vanilla neural network coupled with the dimensionality reduction method (the latest one) showed the apparent order of the dataset; the informative frames are far away from other categories of the uninformative frames. After that, the following idea of clustering, rather than classification, naturally comes to us, as it does not require any labels.

A2 to Q2

We have the comparisons to the state-of-the-art methods, as the Fig. 7, the boxplot of sensitivity for the different methods and ours. Table 4. the classification performance by precision, class-specific recall, and F1-score for comparisons. Other tables or figures show the different combination comparisons in the scheme.

Since our hypothesis differs from others, as we answered the last question, however, based on the performance, we still have the confidence to compare with the state-of-the-art supervised learning methods using the same metrics, such as the precision, sensitivity, and F1-score.

A3 to Q3

For the repeat of the results and the proposed scheme, we prepared a notebook for the implementation of it, as shown in the GitHub repository1.

The results seem like a specific model, but we figured out a scheme; we prefer to name it so. Different clustering methods can be combined with the newly proposed dimensionality reduction technique for a generalisation concern in that scheme.

And still not sure whether to do this at this time, as we introduced as much possible as the four parts construct the scheme (Fig. 1). In different sections, we describe it carefully. As the vanilla neural network structure figure shows in Fig.2. Currently, there is no figure for the dimensionality reduction methods, as we compared three different methods. However, we used our best to describe it in the mathematic language (related to Fig. 4). Later, the chapter on the clustering methods is illustrated. Finally, in the evaluation step, including two algorithms, each of them, we employed the algorithm box to display. We hope such an explanation can dissolve your confusion.

A4 to Q4

Yes, we tried another searching algorithm to optimize such a setting, compared with the random search (section 4.4). In practice, we still try the grid search for the hyperparameter tuning.

The scheme has two kinds of coefficients: the UMAP algorithm and the clustering methods. We employed the Bayesian optimization search to set the optimal one for the parameter space constructed by the two kinds of algorithms.

The purpose of the hyperparameter tuning is to find the optimal coefficients, which leads to the best classification performance, as we used sensitivity as the first priority and precision as the second priority. In other words, the classification metrics construct the cost function to find the optimal coefficients. To the best of our knowledge, the Bayesian optimization search is currently the best one for doing this.

Following your suggestion for novelty, we updated the hypothesis as explained here in the Related Work. Really thanks for your good question to help us re-thinking again and make our work better to understand.

Reference

1. https://github.com/portgasray/NCU-AutoHyp

Round 2

Reviewer 2 Report

ok